# Pretreatment Effect of Inflammatory Stimuli and Characteristics of Tryptophan Transport on Brain Capillary Endothelial (TR-BBB) and Motor Neuron Like (NSC-34) Cell Lines

**DOI:** 10.3390/biomedicines9010009

**Published:** 2020-12-24

**Authors:** Asmita Gyawali, Young-Sook Kang

**Affiliations:** Drug Information Research Institute, College of Pharmacy, Sookmyung Women’s University, Cheongpa-ro 47-gil 100 (Cheongpa-dong 2ga), Yongsan-gu, Seoul 04310, Korea; gyawaliashu@sookmyung.ac.kr

**Keywords:** tryptophan, blood-brain barrier, l-type amino acid transporter 1 system, amyotrophic lateral sclerosis, neurodegenerative disease, motor neuron-like cell lines

## Abstract

Tryptophan plays a key role in several neurological and psychiatric disorders. In this study, we investigated the transport mechanisms of tryptophan in brain capillary endothelial (TR-BBB) cell lines and motor neuron-like (NSC-34) cell lines. The uptake of [^3^H]l-tryptophan was stereospecific, and concentration- and sodium-dependent in TR-BBB cell lines. Transporter inhibitors and several neuroprotective drugs inhibited [^3^H]l-tryptophan uptake by TR-BBB cell lines. Gabapentin and baclofen exerted a competitive inhibitory effect on [^3^H]l-tryptophan uptake. Additionally, l-tryptophan uptake was time- and concentration-dependent in both NSC-34 wild type (WT) and mutant type (MT) cell lines, with a lower transporter affinity and higher capacity in MT than in WT cell lines. Gene knockdown of LAT1 (l-type amino acid transporter 1) and CAT1 (cationic amino acid transporter 1) demonstrated that LAT1 is primarily involved in the transport of [^3^H]l-tryptophan in both TR-BBB and NSC-34 cell lines. In addition, tryptophan uptake was increased by TR-BBB cell lines but decreased by NSC-34 cell lines after pro-inflammatory cytokine pre-treatment. However, treatment with neuroprotective drugs ameliorated tryptophan uptake by NSC-34 cell lines after inflammatory cytokines pretreatment. The tryptophan transport system may provide a therapeutic target for treating or preventing neurodegenerative diseases.

## 1. Introduction

Tryptophan, an essential amino acid for protein biosynthesis, is converted into bioactive metabolites such as serotonin, melatonin, and kynurenine to regulate several physiological processes [1,2]. Serotonin is a neurotransmitter involved in sensory and motor functions and the regulation of emotions [3]. The kynurenine pathway (KP) converts tryptophan to kynurenine with the enzymes indoleamine 2,3-dioxygenase (IDO) and tryptophan 2,3-dioxygenase (TDO), which are involved in the development and prevention of inflammation in the brain [3]. An increase in IDO activity alters serotonin production, leading to the production of kynurenine and neuroactive catabolites, which are either neuroprotective or neurotoxic [4]. Inflammatory cytokines, specifically interferon-γ (IFN-γ), and others, such as tumor necrosis factor-alpha (TNF-α), and free radicals also alter IDO levels [2,5,6]. KP byproduct such as quinolinic acid (an agonist of the N-methyl- d-aspartate (NMDA) receptor) cause neuronal dysfunction; an elevated level of quinolinic acid is indicative of neurodegenerative diseases (NDs) such as Alzheimer’s disease (AD), Parkinson’s disease (PD), Huntington’s disease, autoimmune disease, psychiatric disease, epilepsy, and other infectious diseases [7,8]. However, another byproduct of this pathway is kynurenic acid, an NMDA antagonist, which helps prevent neuronal damage [6,9,10]. Therefore, balanced production of kynurenic and quinolinic acids is important for the prevention of neurodegenerative and psychiatric disorders.

KP metabolites (KPM) are used as diagnostic markers for the progression, severity, and prognosis of diseases of the nervous system [11]. KPM levels are dysregulated in neurodegenerative diseases (ND), including amyotrophic lateral sclerosis (ALS) [12]. ALS is a multifactorial ND characterized by the depletion of neuronal function in the spinal cord and brain. The cerebral spinal fluid and plasma levels of tryptophan are both depleted in ALS patients. Several types of ND, such as AD, PD, and ALS share similar pathological, genetic, and molecular mechanisms of disease manifestation [13,14]. In addition, altered levels of cytokines induced by lipopolysaccharides (LPS) and free radicals, such as TNF-α and IFN-γ, are involved in the induction of oxidative stress and are also associated with the pathophysiology of NDs. Blood-brain barrier (BBB) dysfunction leads to dysregulation of ions, altered homeostasis signaling, and introduction of molecules and toxins into the CNS, leading to homeostatic imbalances and neurodegeneration [15]. Tight junctions exist between the endothelial cells of the BBB, which restrict para-cellular movement and diffusion of solutes in the blood. This restriction protects neural tissues from toxins and pathogens. The transport system across the BBB is classified as, either influx, mediating the transfer of solutes from the blood to the brain, or efflux, mediating the transfer of solutes from the brain to the blood [16]. Up to 75% of albumin-bound tryptophan can cross the BBB [17].

The l-type neutral amino acid transporter 1 (LAT1, slc7a5) is an essential amino acid transporter in the BBB and in ALS cell lines [18,19]. LAT1 is highly expressed in the BBB and in neurons, astrocytes and microglia, and epithelial, testis, placenta, and dendritic cells, transporting phenylalanine, histidine, leucine, tyrosine gabapentin, levodopa, and tryptophan [2,5,19,20,21]. The overexpression of LAT1 has been linked to human cancer development and progression, and its inhibition is a promising therapeutic avenue for cancer treatment [22]. However, low expression of LAT1 in the BBB is also linked to the onset and development of ND-like PD [23]. The role of LAT1 in the brain is to transport tryptophan, which is necessary for neuronal development [19]. According to previous reports, levels of large neutral amino acids (LNAAs), such as citrulline and tryptophan (LAT1), were altered in the ALS mouse model (hSOD1^G93A^). These altered LNAA concentrations were related to BBB transport in the inflammatory state, and direct cellular uptake in the ALS model [24,25]. However, the impact of oxidative stress on LAT1 transporter expression in the BBB in the inflammatory state and on the tryptophan transport mechanism in motor neurons in the ALS model remained unclear.

In this study, we hypothesized that ND pathology is correlated with the inflammatory state and oxidative stress, and that some common neuroprotective drugs may be effective in the prevention of ND. Therefore, we examined the transport characteristics of [^3^H]l-tryptophan under inflammatory conditions in both brain capillary endothelial cell lines, an in vitro model of the BBB, and motor neuron-like cell lines, a cellular model of ALS. Additionally, we aimed to examine the compensatory effects of neuroprotective drugs on both cell types under inflammatory conditions, thus, providing insight into a potential therapy for the prevention and treatment of ND.

## 2. Materials and Methods

### 2.1. Materials

Radiolabeled [^3^H]l-tryptophan (18.4 Ci/mmol) was purchased from PerkinElmer (Waltham, MA, USA). Gabapentin, 3,4-dihydroxy-l-phenylalanine (l-dopa), baclofen, verapamil, phenylalanine, tyrosine, citrulline, glutamine, valine, cysteine, lysine, 2-aminobicyclo heptane-2-carboxylic acid (BCH), harmaline, α-(methylamino)isobutyric acid (MeAIB), homoarginine, dopamine, donepezil, riluzole, and amiloride were obtained from Sigma-Aldrich (St. Louis, MO, USA) and Tokyo Kasei Kogyo Co. (Tokyo, Japan). All chemicals and reagents were of reagent grade.

### 2.2. Cell Culture

Brain capillary endothelial (TR-BBB, passages 22–28) and motor neuron-like NSC-34 (wild type [WT]: NSC-34/hSOD1^wt^ and mutant type [MT]: NSC-34/hSOD1^G93A^; passages 10–19) cell lines were cultured using protocols described in our previous studies [24,26]. TR-BBB and NSC-34 cell lines were obtained from Prof. Tetsuya Terasaki (Tohoku University, Sendai, Japan), and Prof. Hoon Ryu (KIST, Seoul, Korea), respectively. Cells were cultured in DMEM (Dulbecco’s modified Eagle’s medium) (Invitrogen, San Diego, CA, USA) supplemented with 10% fetal bovine serum, 100 U/mL penicillin, and 100 µg/mL streptomycin (Invitrogen). For TR-BBB and NSC-34 cell lines, initial seeding (1 × 10^5^ cells/well, and 3 × 10^5^ cells/well, respectively) was carried out on rat tail collagen type I-coated 24- and 6-well plates (IWAKI, Tokyo, Japan), followed by 48-h incubation at 33 °C and 37 °C, respectively [25,27].

### 2.3. [^3^H]l-Tryptophan Uptake by NSC-34 and TR-BBB Cells

To investigate [^3^H]l-tryptophan uptake by TR-BBB and NSC-34 cell lines, we followed a previously published protocol [28]. We prepared the extracellular fluid (ECF) buffer (pH 7.4), as previously described [29]. [^3^H]l-Tryptophan (136 nM) uptake by TR-BBB and NSC-34 cell lines was measured in ECF buffer (pH 7.4) in the presence or absence of unlabeled inhibitors (0.5–2 mM) incubated at 37 °C for the designated time [24]. To evaluate sodium dependency, NaCl was replaced with N-methyl- d-glucamine (NMG), and NaHCO_3_ was replaced with KHCO_3_ in the ECF buffer. Similarly, to investigate chloride dependency, NaCl, KCl, and CaCl_2_ in the ECF buffer were replaced with sodium, potassium, and calcium gluconate, respectively [28]. After the uptake analysis, 750 µL of 1 N NaOH and phosphate-buffered saline (PBS) were used to dissolve the cells overnight. The next day, radioactivity was measured using an LS6500 liquid scintillation counter (Beckman, Fullerton, CA, USA). In addition, the cell-to-medium ratio (µL/mg protein) was calculated using the following equation:(1)$$\mathrm{Cell}/\mathrm{medium}\;\mathrm{ratio}\;=\frac{\left[3H\right]\mathrm{dpm}\;\mathrm{per}\;\mathrm{cell}\;\mathrm{protein}\left(\mathrm{mg}\right)}{\left[3H\right]\mathrm{dpm}\;\mathrm{per}\;µL\;\mathrm{medium}}.$$

### 2.4. Transfection with rLAT1 and rCAT1 Small Interfering RNA (siRNA)

For transfection and knockdown analysis, 200 nM LAT1 and cationic amino acid transporter 1 (CAT1, slc7a1) siRNA or negative control siRNA (control pool) (GE Healthcare Dharmacon, Inc., Lafayette, CO, USA) were transfected into NSC-34 and TR-BBB cell lines using Lipofectamine^®^ 2000 (Invitrogen, Carlsbad, CA, USA). For RNA interference analysis, total RNA (2 µg) was isolated from NSC-34 and TR-BBB cell lines using the RNeasy kit (Qiagen, Valencia, CA, USA) according to the manufacturer’s instructions. RNA was reverse transcribed using a High-Capacity cDNA Reverse Transcription Kit (Applied Biosystems, Life Technologies, Foster City, CA, USA). After 48 h and 24 h, the transfected TR-BBB, and NSC-34 cell lines, respectively, were used for [^3^H]l-tryptophan uptake analysis, and relative gene expression was measured using GAPDH as a control. Real-time PCR was performed in 48-well plates using the StepOne Real Time-PCR system (Applied Biosystems, Life Technologies) [24,27]. Similarly, LAT1 mRNA expression was assessed after 24-h pretreatment with LPS and TNF-α [30].

### 2.5. Pretreatment Effect of Pro-Inflammatory Cytokines on the Uptake of [^3^H]l-Tryptophan

TR-BBB and NSC-34 cell lines were pre-treated with LPS (20 ng/mL) and TNF-α (20 ng/mL) for 24 h [29,30,31], followed by [^3^H]l-tryptophan uptake analysis. Additionally, TNF-α/LPS-treated TR-BBB and NSC-34 cell lines were incubated for 24 h with LAT1 substrates/inhibitors or neuroprotective drugs, including baclofen (2 mM), gabapentin (0.5 mM), l-dopa (0.1 mM), and verapamil (0.1 mM), prior to [^3^H]l-tryptophan uptake analysis.

### 2.6. Data Analysis

For kinetic studies, Sigma plot ver. 12 (Sigma-Aldrich) was used for non-linear least-squares regression analysis. The concentration (C), Michaelis-Menten constant (K_m_), maximum uptake rate (V_max_), and first-order constant of the non-carrier-mediated process (K_d_) of [^3^H]l-tryptophan were calculated using the following equation,
V = V_max_ × C/(K_m_ + C) + K_d_ × C(2)
where V is the initial uptake rate of [^3^H]l-tryptophan at 5 min, C is the concentration of l-tryptophan. The initial uptake rate of [^3^H]l-tryptophan was estimated in the presence of compounds such as gabapentin and baclofen, using the following equation,
V = V_max_ × C/[K_m_ × (1 + I/K_i_) +C] + K_d_ × C(3)
where, I represents the concentration of the compounds used for kinetic inhibition (K_i_) and K_i_ is the inhibition constant. Statistical differences were calculated using the Student’s *t*-test and one-way ANOVA with Dunnett’s test, post hoc.

## 3. Results

### 3.1. Characteristics of [^3^H]l-Tryptophan Transport in TR-BBB Cell Lines

We used TR-BBB cell lines as an in vitro system to characterize the transport mechanism of tryptophan across the BBB. Tryptophan uptake was carried out in the presence or absence of unlabeled l-tryptophan and d-tryptophan at a concentration of 2 mM at pH 7.4 for 5 min. [^3^H]l-Tryptophan uptake by TR-BBB cell lines was significantly inhibited by 90% and 65% in the presence of l-and d-tryptophan, respectively (Figure 1A). According to the stereospecificity demonstrated, a concentration range of 0–2 mM unlabeled l-tryptophan was selected for the concentration dependency study using TR-BBB cell lines. A Michaelis-Menten constant (K_m_) of 121 ± 11 µM, maximum uptake rate (V_max_) of 4.69 ± 0.20 nmol/mg protein/min, and a K_d_ value of 3.66 µL/mg protein/min were obtained for [^3^H]l-tryptophan uptake by TR-BBB cell lines (Table 1). The Scatchard plot followed a straight line, indicating a saturable transport system for the uptake of [^3^H]l-tryptophan (Figure 1B). In addition, replacement of Na^+^ in the ECF buffer with NMG^+^ resulted in a significant reduction in tryptophan uptake, indicating sodium dependency. However, replacement of Cl^-^ in the ECF buffer with sodium gluconate did not induce a significant differently [^3^H]l-tryptophan uptake (Figure 1C). These results indicated that l-tryptophan uptake was time, concentration-, and sodium-dependent, with a high-affinity and low-capacity transport system existing in TR-BBB cell lines.

To examine the effects of several transporters, we selected various transporter substrate/inhibitor concentrations depending on the obtained K_m_ value for [^3^H]l-tryptophan uptake by TR-BBB cell lines for 5 min at pH 7.4 (Table 2). [^3^H]l-Tryptophan uptake was inhibited by >70% in the presence of system L inhibitors, such as BCH and d-methionine. An inhibitor of system y^+^ L, l-methylmaleimide, also inhibited tryptophan uptake by 40%. However, inhibitors of systems y^+^ (homoarginine), ATA2 (MeAIB), and b°^+^ (harmaline) did not affect tryptophan uptake (Table 2). These results confirmed that system L played a greater role in the uptake of tryptophan than other systems. Next, we investigated the effect of several amino acids on the uptake of [^3^H]l-tryptophan by TR-BBB cell lines (Table 3). l-Type amino acids (LAT1 substrates) such as phenylalanine, tyrosine, methionine, and citrulline exerted a strong inhibitory effect (>70%). Other l-type amino acids such as glutamine, valine, and cysteine induced >60% inhibition, while lysine induced >40% inhibition on tryptophan uptake, likely due to stereospecific effects. LAT1 is mostly stereoselective for l-type amino acids. In contrast, acidic amino acids such as aspartic and glutamic acids, and other amino acids such as taurine and choline, had no effect on tryptophan uptake (Table 3). These results indicated that neutral amino acids (NAAs) significantly inhibited the uptake of [^3^H]l-tryptophan, suggesting their involvement in the transport of tryptophan in TR-BBB cell lines.

After the inhibition analog study, the inhibition kinetics of [^3^H]l-tryptophan uptake by TR-BBB cell lines were examined using gabapentin (400 µM) and baclofen (2 mM) (Figure 2). The Lineweaver-Burk plot demonstrated competitive inhibition in the presence or absence of gabapentin and baclofen with K_i_ values of 0.21 mM, and 1.62 mM, respectively (Figure 2).

### 3.2. LAT1 and CAT1 siRNA Transfection Study in TR-BBB Cell Lines

To examine the contribution of LAT1 and CAT1 transporters to the transport of l-tryptophan, TR-BBB cell lines were transfected with siRNA to knock down LAT1 and CAT1 expression. The relative mRNA expression of LAT1 and CAT1 was suppressed, confirming the presence of both transporters in TR-BBB cell lines (Figure 3A). Tryptophan uptake by the siRNA-transfected cells was then measured (Figure 3B), revealing that only LAT1 played a major role in the transport of [^3^H]l-tryptophan in TR-BBB cell lines.

### 3.3. Effect of Pharmacological Drugs on the Uptake of [^3^H]l-Tryptophan by TR-BBB Cell Lines

The transport mechanism of tryptophan in the BBB was investigated using several pharmacological drugs (Table 4). Verapamil, a Ca^2+^ antagonist used for the treatment of hypertension, l-dopa, and gabapentin are the substrates of system L and strongly inhibit the uptake of tryptophan. Likewise, baclofen and quinidine exerted a small inhibitory effect (~21%) on [^3^H]l-tryptophan uptake. However, no effect on tryptophan uptake was observed in the presence of CNS-acting drugs such as dopamine, donepezil, riluzole, and amiloride, (Table 4).

### 3.4. Pretreatment Effect of Inflammatory Stimuli on [^3^H]l-Tryptophan Uptake by TR-BBB Cell Lines

Changes in [^3^H]l-tryptophan uptake by TR-BBB cell lines were evaluated after 24 h pretreatment with inflammatory stimuli TNF-α (20 ng/mL) and LPS (20 ng/mL) (Figure 4). Pretreatment of TR-BBB cell lines with TNF-α and LPS significantly increased [^3^H]l-tryptophan uptake by 133%, and 143%, respectively, compared to the control (Figure 4). To examine the anti-inflammatory effects of the neuroprotective drugs baclofen, gabapentin, l-dopa, and verapamil, TR-BBB cell lines were then incubated with the neuroprotective drugs for 24 h. Tryptophan uptake remained significantly increased in the presence or absence of inflammatory stimuli, demonstrating that the neuroprotective drugs mitigated the effects of the inflammatory stimuli and protected endothelial cells from inflammation.

### 3.5. Changes in [^3^H]l-Tryptophan Uptake by NSC-34 Cell Lines.

To identify changes in [^3^H]l-tryptophan uptake by NSC-34 WT and MT cell lines, we conducted time- and concentration-dependent uptake assays. The time dependency uptake assay was carried out from 5 s to 60 min, revealing a linear uptake until 5 min in both WT and MT cell lines (Figure 5A). After 5 min, the uptake of tryptophan by MT cell lines was significantly lower than that by WT cell lines; therefore, we selected a 5-min study duration for further experiments. The concentration dependency uptake analysis in WT and MT cell lines determined a K_m_ of 33.1 ± 1.2 µM and 101 ± 2 µM, and V_max_ of 2.10 ± 0.27 and 4.07 ± 0.42 nmol/mg protein/min, respectively. The Scatchard plot followed a straight line, indicating a saturable transport mechanism for the uptake of tryptophan (Figure 5B embedded graph). These results indicated that l-tryptophan uptake was relatively lower in MT cell lines than in WT cell lines, and was time- and concentration-dependent, with a lower affinity and higher capacity transport system.

### 3.6. Effect of LAT1 siRNA Transfection on NSC-34 Cell Lines

LAT1 knockdown by siRNA transfection in NSC-34 cell lines was carried out to determine LAT1 involvement in the transport of l-tryptophan in NSC-34 WT and MT cell lines. SiRNA-transfected NSC-34 WT and MT cell lines were used to assess the mRNA expression level and uptake. The LAT1 mRNA expression and tryptophan uptake results indicated significant depletion effect in LAT1 siRNA-transfected WT and MT cell lines (Figure 5). Additionally, both LAT1 mRNA expression and tryptophan uptake were decreased in MT cell lines compared to WT cell lines, demonstrating consistent results with the time dependency results (Figure 5 and Figure 6).

### 3.7. Pretreatment Effect of Inflammatory Stimuli on [^3^H]l-Tryptophan Uptake by NSC-34 Cell Lines

To identify alterations in [^3^H]l-tryptophan transport and transporter expression in the inflammatory state, pretreatment was performed for 24 h using oxidative stress-inducing agents TNF-α (20 ng/mL) and LPS (20 ng/mL) on NSC-34 WT and MT cell lines. To confirm the potential anti-inflammatory effects of neuroprotective drugs, NSC-34 cell lines (WT and MT) were then incubated with baclofen, gabapentin, l-dopa, and verapamil for 24 h. The tryptophan uptake was significantly decreased by 20–25% after pretreatment with TNF-α and LPS compared to that in the non-treated control (Figure 7A–D). However, the uptake rate of l-tryptophan was restored by treatment with neuroprotective drugs (Figure 7A–D), indicating that the neuroprotective drugs exerted anti-inflammatory effects in the NSC-34 cell lines. To confirm the transporter expression pattern under inflammatory conditions, we measured the mRNA expression levels of the LAT1 transporter, which were significantly decreased (Figure 7E,F). [^3^H]l-Tryptophan uptake and LAT1 mRNA expression in the NSC-34 cell lines under inflammatory conditions were consistent (Figure 7).

## 4. Discussion

Tryptophan, an amino acid precursor of the neurotransmitter serotonin (5-HT), is implicated in a range of brain functions. Brain tryptophan concentration is correlated with that of serotonin and 5-hydroxyindoleacetic acid (5-HIAA), a major serotonin metabolite [8]. Tryptophan is transported into the serotonergic neuron via a high-affinity transporter system, which is necessary for normal neuronal development in the brain [8,19]. Therefore, to clarify the transport mechanism of tryptophan in endothelial and neuronal cell lines, cellular uptake of [^3^H]l-tryptophan and mRNA expression levels of tryptophan transporters were analyzed.

Our results demonstrate that [^3^H]l-tryptophan uptake by TR-BBB cells is stereospecific and dependent on sodium and tryptophan concentration, indicating a single saturable high affinity and low capacity transport system (Figure 1). The affinity and capacity of tryptophan uptake were dependent on previously referenced concentration ranges [32]. According to previously published results, the transport of NAAs by the LAT1 system is sodium-independent whereas that of basic amino acids is sodium-dependent [33]. Based on our results and the published literature, it can be concluded that tryptophan uptake is sodium-dependent in TR-BBB and fibroblast cell lines [32,33,34]. In the brain, uptake of NAAs is mediated through the LAT1 system [18]. Consequently, known LAT1 inhibitors such as BCH, d-methionine, and N-methylmaleimide (NMM) inhibited the cellular uptake of [^3^H]l-tryptophan (Table 2) [35]. Schizophrenia, neuropsychiatric disorders, and PD are largely dependent upon the brain uptake of LNAAs such as tryptophan and tyrosine. The amino acids tryptophan and tyrosine are precursors for serotonin, and dopamine/epinephrine, respectively. Therefore, LAT1 is believed to play a major role in maintaining brain homeostasis through the regulation of neurotransmitter synthesis [34,36]. l-Tryptophan uptake was significantly inhibited by NAAs in TR-BBB cell lines, but no effect was observed in the human macrophage cell lines [5,37]. Therefore, NAAs may have a major impact on the transport of tryptophan in the BBB. In contrast, inhibitors of systems y^+^, A, and b°^+^, such as homoarginine, MeAIB, and harmaline, respectively, had no impact on tryptophan uptake. As negative controls, taurine (transported through the BBB by TAUT, the taurine transporter), choline (transported by CHT, the choline transporter), and glutamic and aspartic acids (acidic amino acids) had no effect on tryptophan uptake by TR-BBB cell lines [26,29]. The current inhibition results demonstrate that tryptophan is transported to the brain through the LAT1 transporter (Table 3). Gabapentin and baclofen, structural analogs of gamma-aminobutyric acid (GABA) and substrates of LAT1, showed a competitive inhibition effect on [^3^H]l-tryptophan uptake by TR-BBB cell lines (Figure 2) [19,33].

Several drugs, used to treat CNS disorders, were investigated their effect on the [^3^H]l-tryptophan uptake in this study (Table 4). Verapamil (l-type calcium channel blocker with high BBB permeability that exerts neuroprotective effects against several neurological disorders) inhibited tryptophan uptake by TR-BBB cell lines [38]. Both levodopa (l-dopa) and gabapentin (therapeutic drugs for PD and chronic neuropathic pain) also strongly inhibited tryptophan uptake [35,39,40]. Similarly, a low dosage of baclofen (a selective GABA-B receptor agonist and neuroprotective therapeutic used to improve learning/memory function in patients with AD by normalizing protein expression in the brain) inhibited tryptophan uptake [41]. However, dopamine (a neurotransmitter and therapeutic for PD) and donepezil (CHT inhibitor used to treat AD) did not show any effect on tryptophan uptake [26,39]. Similarly, riluzole (therapeutic for ALS) and amiloride possess neuroprotective properties for CNS injury and neuropathological conditions but had no effect on tryptophan uptake by TR-BBB cell lines (Table 4) [42].

Tryptophan uptake was time- and concentration-dependent in NSC-34 cell lines. The time-dependent uptake was decreased in MT cell lines compared to that in WT cell lines. In the [^3^H]l-tryptophan uptake kinetic analysis, the transport system in MT cell lines had lower affinity with less sensitivity and higher capacity than that in WT cell lines (Figure 5 and Table 1). According to previous studies, the LAT1 transporter system is highly expressed in the brain capillary cells [43], the spinal dorsal horn [35], primary neurons, astrocytes, and microglia [21] and transport NAA, whereas CAT1 transports basic amino acids such as l-arginine, l-lysine, and l-ornithine at the BBB [44]. Therefore, downregulation of LAT1 expression in TR-BBB and NSC-34 cell lines was achieved via siRNA silencing, and the subsequent uptake of [^3^H]l-tryptophan was measured. The results indicated that primarily, LAT1 is involved in the transport of tryptophan in both BBB and NSC-34 cell lines (Figure 3 and Figure 6). Tryptophan uptake was not affected by CAT1 siRNA transfection in TR-BBB cell lines, suggesting that CAT1 is not involved in tryptophan transport. However, both LAT1 mRNA expression and tryptophan uptake were decreased in MT cell lines compared to WT cell lines, suggesting that the mutant model cell line transporters were likely deleted or mutated.

The effect of pro-inflammatory cytokine (LPS and TNF-α) pretreatment for 24 h on TR-BBB and NSC-34 cell lines was determined by measuring [^3^H]l-tryptophan uptake. [^3^H]l-Tryptophan uptake by TR-BBB cell lines increased after incubation with the inflammatory stimuli, which further increased with the addition of neuroprotective drugs (Figure 4). The elevated influx of [^3^H]l-tryptophan transport might have been the result of alterations in the transport mechanisms due to induction of free radicals and oxidative stress by the pro-inflammatory cytokines [29] (Figure 4A,B). An increase in oxidative stress triggers an increased production of pro-inflammatory cytokines, which affect the expression of lipid membrane components and the transport function of tryptophan in the BBB [45]. On the other hand, inflammatory stimuli-treated NSC-34 WT and MT cell lines demonstrated significantly reduced tryptophan uptake (Figure 7). An excessive amount of TNF-α in neuronal cells can produce excessive extracellular glutamate levels and reduce glutamate clearance, whereas LPS treatment activates the production of intracellular molecules that further activate the production of inflammatory mediators, key factors in motor neuron damage, and diminishes the uptake of tryptophan in motor neurons [46,47]. Up- and down-regulation of transporters expression in the inflammatory state depended on the cell type; we observed similar results to those of a previous study with motor neuronal and endothelial cell lines [29,30]. Along with inflammatory stimuli, neuroprotective drugs such as baclofen, gabapentin, l-dopa, and verapamil restored tryptophan uptake by NSC-34 cell lines (Figure 7). Baclofen, an anti-inflammatory GABA receptor agonist, has a diminished effect against LPS-induced toxicity and improves neuronal survival in NDs [41]. Similarly, gabapentin, an anti-glutamatergic drug, decreases glutamate excitotoxicity and prolongs motor neuronal survival [48]. l-Dopa is a precursor of dopamine that can cross the BBB; hence, it is useful for prevention of early-stage PD [39]. The neuroprotective mechanism of verapamil in an ALS mouse model (SOD1^G93A^ mice) was suggested to be via the depletion of protein aggregation, which enhanced motor neuronal survival by ameliorating endoplasmic reticulum stress [49]. Verapamil blocks LPS-induced neurotoxicity in dopaminergic neurons by exerting an anti-inflammatory effect; therefore, tryptophan uptake by TR-BBB cell lines increased after verapamil pretreatment [38]. Neuroprotective/anti-inflammatory drugs significantly increase and restore tryptophan uptake; hence, these drugs exerted anti-inflammatory effects in both TR-BBB and NSC-34 cell lines (Figure 4 and Figure 7). Consistently, we found that LAT1 mRNA expression decreased after pretreatment with inflammatory stimuli in NSC-34 cell lines, suggesting that LAT1 transporter expression decreased under inflammatory conditions; therefore, tryptophan uptake was reduced (Figure 7). We hypothesize that pretreatment with pro-inflammatory cytokines induces IDO production in the KP, which affects the signaling pathway and activates the NMDA receptor agonist, ultimately causing neuro-inflammation/oxidative stress in the cell lines. Hence, the aforementioned mechanism might suppress LAT1 expression in the inflammatory state.

## 5. Conclusions

Tryptophan is an amino acid precursor of serotonin, implicated in neuropsychiatric conditions such as depressive disorders, schizophrenia, and autism. The uptake of tryptophan was shown to be affected by kinetic parameters, its concentration, sodium, structural analogs, and inhibitors. Our findings demonstrated that tryptophan transport in TR-BBB and NSC-34 cell lines were primarily facilitated by the LAT1 transporter system, which was altered in NSC-34 cell lines. Under inflammatory conditions, treatment with neuroprotective/anti-inflammatory drugs attenuated inflammation in both cell types. We believe that the transport mechanisms of tryptophan could provide useful biomarkers for the target, design, and development of new therapeutic and preventative strategies for neurodegenerative and neuro-inflammatory disorders. Targeting the tryptophan transport system through the brain and neuronal cells may improve the efficacy and safety of therapeutics in NDs.

## Figures and Tables

**Figure 1 biomedicines-09-00009-f001:**
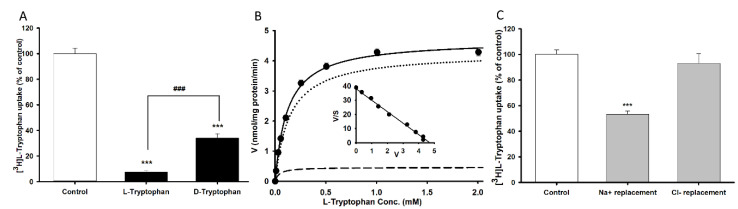
Stereospecificity, concentration, and ion dependency of [^3^H]l-tryptophan uptake by tryptophan in brain capillary endothelial (TR-BBB) cell lines at 37 °C for 5 min at pH 7.4. (**A**) [^3^H]l-Tryptophan uptake in the presence of 2 mM unlabeled l- and d-tryptophan. (**B**) Concentration-dependent tryptophan uptake with 0–2 mM unlabeled l-tryptophan. The embedded graph shows the Scatchard plot. The line with dots represents total uptake rate, dot line represents carrier mediated transport and line of dash denotes non-specific transport of tryptophan. (**C**) [^3^H]l-Tryptophan uptake when sodium was replaced with NMG and chloride was replaced with gluconate in the transport buffer. Each point represents the mean ± SEM (*n* = 3–4). *** *p* < 0.001 indicates a significant difference from the control, ^###^
*p* < 0.001 indicates a significant difference from l-tryptophan.

**Figure 2 biomedicines-09-00009-f002:**
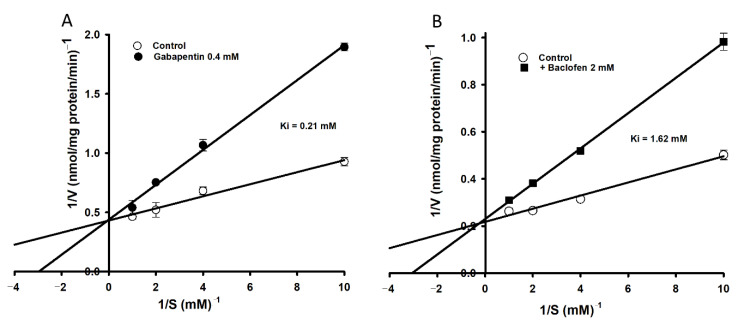
Lineweaver-Burk plots of [^3^H]l-tryptophan uptake by TR-BBB cell lines showing competitive inhibition. The uptake was performed at 37 °C for 5 min in the presence (●/■) or absence (○) of (**A**) gabapentin (0.4 mM, ●) and (**B**) baclofen (2 mM, ■). Each point represents the mean ± SEM (*n* = 3).

**Figure 3 biomedicines-09-00009-f003:**
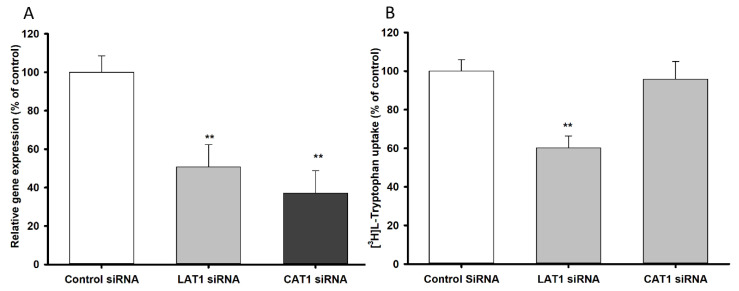
LAT1 and CAT1 siRNA knockdown effect on (**A**) mRNA expression and (**B**) [^3^H]l-tryptophan uptake by TR-BBB cell lines. The siRNA transfection was carried out for 48 h, followed by mRNA expression and uptake study at 37 °C for 5 min. Each point represents the mean ± SEM (*n* = 4). ** *p* < 0.01 indicates a significant difference from the control.

**Figure 4 biomedicines-09-00009-f004:**
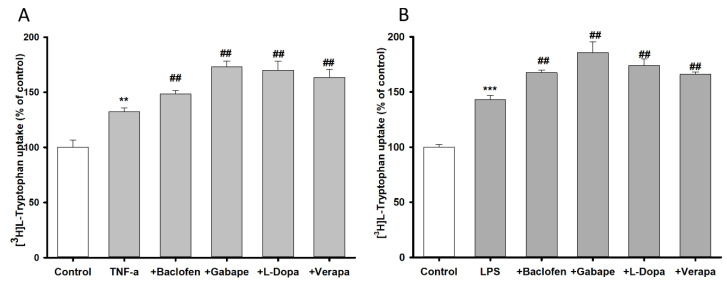
Pretreatment effect of inflammatory stimuli on [^3^H]l-tryptophan uptake by TR-BBB cell lines at 37 °C for 5 min. (**A**) TNF-α and (**B**) LPS were used at a concentration of 20 ng/mL, followed by treatments with baclofen (2 mM), gabapentin (0.5 mM), l-dopa (0.1 mM), and verapamil (0.1 mM). Data are presented the mean ± SEM (*n* = 4). *** *p* < 0.001, ** *p* < 0.01 indicates a significant difference from the control, ^##^
*p* < 0.01 indicates a significant difference from the respective control (TNF-α or LPS).

**Figure 5 biomedicines-09-00009-f005:**
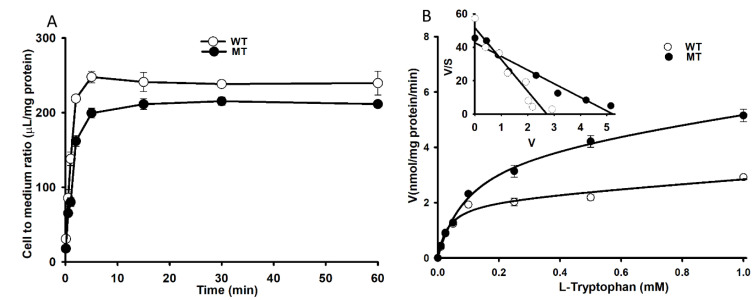
Time and concentration dependency of [^3^H]l-tryptophan uptake by NSC-34 cell lines for 5 min at pH 7.4 (**A**) Time dependent uptake was evaluated between 5 s to 60 min. (**B**) Concentration dependency was evaluated for 5 min in the presence of unlabeled l-tryptophan (0–1 mM). The embedded graph shows the Scatchard plot in NSC-34 cell lines. (○) represents NSC-34 wild type cell lines (WT, NSC-34 hSOD1^WT^); (●) represents NSC-34 mutant type (MT, NSC-34 hSOD1^G93A^) cell lines. Each point represents the mean ± SEM (*n* = 3).

**Figure 6 biomedicines-09-00009-f006:**
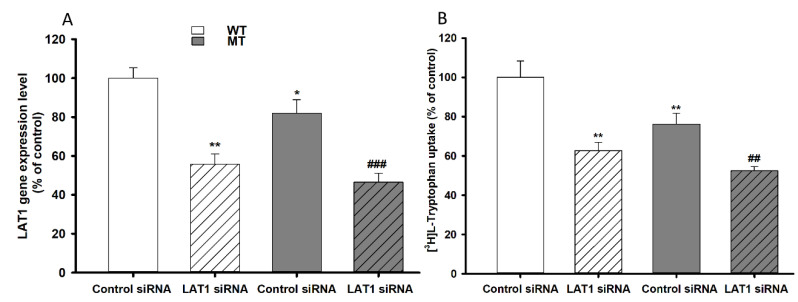
Effects of LAT1 knockdown measured by (**A**) LAT1 mRNA expression and (**B**) [^3^H]l-tryptophan uptake by transfected NSC-34 cell lines The siRNA transfection was carried out for 24 h, then the mRNA expression and uptake analysis were carried out at 37 °C for 5 min. ** *p* < 0.01, * *p* < 0.05 indicate a significant difference from the WT control, ^###^
*p* < 0.001, ^##^
*p* < 0.01 indicates a significant difference from the MT control. Each point represents the mean ± SEM (*n* = 3–4). WT, wild-type NSC-34 cell lines; MT, mutant-type NSC-34 cell lines.

**Figure 7 biomedicines-09-00009-f007:**
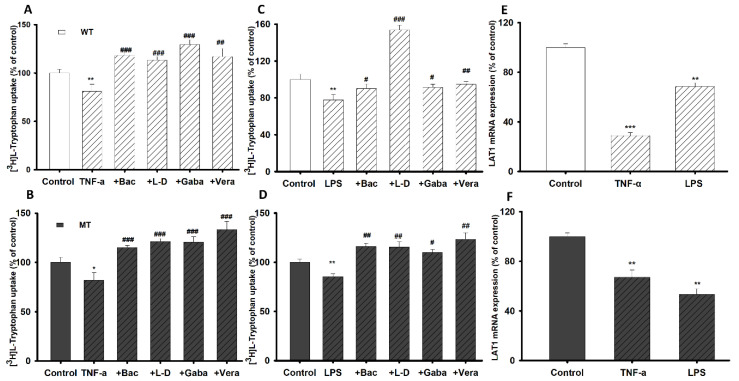
Pretreatment effect of inflammatory stimuli on NSC-34 cell lines in the presence or absence of neuroprotective drugs (Bac; baclofen, l-d; l-dopa, Gaba; gabapentin, and Vera; verapamil). (**A**) Effect of TNF-α on tryptophan uptake on NSC-34 (**A**) WT cell lines and (**B**) MT cell lines. Effect of LPS on tryptophan uptake on NSC-34 (**C**) WT and (**D**) MT cell lines. (**E**,**F**) LAT1 mRNA expression assessed by quantitative real-time PCR analysis, normalized to GAPDH. Data are presented as the mean ± SEM (*n* = 4). *******
*p* < 0.001, ******
*p* < 0.01, *****
*p* < 0.05 indicate a significant difference from the control; ^###^
*p* < 0.001, ^##^
*p* < 0.01, ^#^
*p* < 0.05 indicate a significant difference from the respective control (TNF-α or LPS). WT, wild-type cell lines; MT, mutant-type NSC-34 cell lines.

**Table 1 biomedicines-09-00009-t001:** Kinetic parameters of [^3^H]l-tryptophan uptake by TR-BBB and NSC-34 cell lines.

Cell lines	WT (NSC-34 hSOD1^WT^)	MT (NSC-34 hSOD1^G93A^)	TR-BBB
K_m_ (µM)	33.1 ± 1.2	101 ± 2	121 ± 11
V_max_ (nmol/mg protein/min)	2.10 ± 0.27	4.07 ± 0.42	4.69 ± 0.20
K_d_ (µL/mg protein/min)	0.805	1.46	3.66
V_max_/K_m_ (µL/mg protein/min)	0.063	0.041	0.704

K_m_, transport affinity; V_max_, maximum transport velocity; K_d_, non-saturable transport clearance, V_max_/K_m_, intrinsic transport clearance; WT, wild-type; MT, mutant-type. Each value represents the mean ± SEM.

**Table 2 biomedicines-09-00009-t002:** Effect of transporter inhibitor on [^3^H]l-tryptophan uptake by tryptophan in brain capillary endothelial (TR-BBB) cell lines.

Compounds	Concentration	Relative Uptake
	mM	(% of Control)
Control		100 ± 4
+ BCH	2	21.0 ± 1.3 ***
+ d-Methionine	2	30.8 ± 3.4 ***
+ Methylmaleimide	2	60.4 ± 1.2 ***
+ Homoarginine	2	88.4 ± 3.4
+ MeAIB	2	96.8 ± 2.7
+ Harmaline	2	108 ± 5

The uptake of [^3^H]l-tryptophan was performed at 37 °C for 5 min in TR-BBB cell lines with or without the presence of transporter inhibitors at a concentration of 2 mM. The above-presented values represent the mean ± S.E.M. (*n* = 3–4). *** *p* < 0.001 denotes significantly different from control.

**Table 3 biomedicines-09-00009-t003:** Effect of various compounds on [^3^H]l-tryptophan uptake by TR-BBB cell lines.

Compounds	Concentration	Relative Uptake
	mM	(% of Control)
Control		100 ± 4
+ Tryptophan	2	7.27 ± 0.9 ***
+ Phenylalanine	2	8.68 ± 0.3 ***
+ Tyrosine	2	14.1 ± 1.0 ***
+ Methionine	2	10.3 ± 7.4 ***
+ Glutamine	2	19.4 ± 1.2 ***
+ Citrulline	2	27.3 ± 1.4 ***
+ Valine	2	38.4 ± 1.8 ***
+ Cysteine	2	41.3 ± 1.3 ***
+ Lysine	2	53.1 ± 4.1 ***
+ Serine	2	58.6 ± 1.5 ***
+ Aspartic acid	2	90.2 ± 3.4
+ Glutamic acid	2	97.4 ± 3.8
+ Taurine	2	102 ± 5
+ Choline	2	110 ± 4

[^3^H]l-Tryptophan uptake by TR-BBB cell lines at 37 °C for 5 min in the presence or absence of l-type amino acids, taurine and choline at a concentration of 2 mM. Each value represents the mean ± SEM (*n* = 3–4). *** *p* < 0.001 denotes significantly different from control.

**Table 4 biomedicines-09-00009-t004:** Effect of selected pharmacological drugs on [^3^H]l-tryptophan uptake by TR-BBB cell lines.

Drugs	Concentration	Relative Uptake
	mM	(% of Control)
Control		100 ± 4
+ Verapamil	0.5	15.0 ± 0.9 ***
+ l-Dopa	0.5	18.6 ± 1.6 ***
+ Gabapentin	0.5	46.0 ± 0.5 ***
+ Baclofen	2	73.1 ± 5.3 **
+ Quinidine	0.5	78.5 ± 5 **
+ Dopamine	2	94.4 ± 3.3
+ Donepezil	2	97.1 ± 7.6
+ Riluzole	0.5	102 ± 8
+ Amiloride	2	111 ± 6

The uptake of [^3^H]l-tryptophan was carried out by TR-BBB cell lines at 37 °C for 5 min in pH 7.4, in the presence or absence of drugs. Each value represents the mean ± SEM (*n* = 3–4). *** *p* < 0.001, ** *p* < 0.01 indicates a significant difference from the control.

## Data Availability

Data is contained within the article.

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
