# Peer review of "Pretreatment Effect of Inflammatory Stimuli and Characteristics of Tryptophan Transport on Brain Capillary Endothelial (TR-BBB) and Motor Neuron Like (NSC-34) Cell Lines"

_biomedicines, 2020, doi:10.3390/biomedicines9010009_

Round 1
Reviewer 1 Report
In this manuscript, the transport mechanisms of tryptophan in brain capillary endothelial and motor neuron like cell lines were investigated. Effect of pharmacological compounds and pretreatment effect of inflammatory stimuli on the uptake of tryptophan were comprehensively characterized. Thus, I recommend publishing this paper in this journal.
Author Response
Response: Thank you so much for your review and we appreciate for the reviewer positive response. English language and style has been corrected by native English speaker.
Reviewer 2 Report
The manuscript from Gyawali and Kang describes the characterization of Tryptophan uptake in brain capillary endothelial and motor neuron like cell linses.
The enzyme kinetic experiments are well designed and allow an adequate determination of Km and Vmax.
The manuscript shows, however, several weaknesses:
The inhibitor experiments were performed at only one concentration. For the active compounds IC50 determination would be required. Cytotoxicity as potential cause of some of the effects was not considered. The current Picture is that the some of the amino acides Show a stronger Inhibition of tryptophan Transport than the Transporter Inhibitors investigated. Also Verapamil and L-dopa also not considered LAT1 Inhibitors showed higher Inhibition at lower concentrations compared to the transporter Inhibitors.
The stimulation with LPS and TNF had an only moderate but contrary effects on tryptophan transport in both cell lines. Further more, the inhibitors shown to have substantial inhibitory effects in unstimulated cells did not reduce the tryptophan transport but rather increased it. In light of these overal rather confusing data it is not sufficient to come up with speculations in the discussion. Experimental proof of at least some of these speculations is mandatory to increase the trust into the reported data.
Inhibition with the siRNA unfortunately reached only 50%. Again only one concentration and one pre-incubation time is reported here. Concentration-dependency and variation of preincubation time or preincubation design would have been needed. In addition potential cytotoxicity of the siRNA treatment are missing.
Some minor Details :
Figure 1: Statistic Evaluation L- vs. D-Tryptophan missing, would be needed to support statemten on stereospecific effect on the uptake (line 154-155)
Figure 1: Text for subfigure A and B are mixed up; description of subfigure b is incomplete, meaning of different lines in the plot is missing
Figure 5: Scatchard plot too smal
Figure 6: unit of y-axis missing
Author Response
Reviewer 2 (Round 1)
Comments and Suggestions for Authors
The manuscript from Gyawali and Kang describes the characterization of Tryptophan uptake in brain capillary endothelial and motor neuron like cell lines.
Response: Thank you so much for your review, comments and suggestions.
Our answers have made in the blue color, comments are mentioned in the black color and correction in manuscript are in red color.
English language and style have been corrected by native English speaker.
The enzyme kinetic experiments are well designed and allow an adequate determination of Km and Vmax.
The manuscript shows, however, several weaknesses:
1.The inhibitor experiments were performed at only one concentration. For the active compounds IC50 determination would be required. Cytotoxicity as potential cause of some of the effects was not considered. The current Picture is that the some of the amino acids show a stronger Inhibition of tryptophan Transport than the Transporter Inhibitors investigated. Also Verapamil and L-dopa also not considered LAT1 Inhibitors showed higher Inhibition at lower concentrations compared to the transporter Inhibitors.
Response:
- In L-tryptophan concentration dependency study, we used unlabeled L-tryptophan concentration ranging from 0-2 mM which has shown the self-inhibition with the Km value of 121 ± 11 mM. Usually 10 times higher value of the Km can be used for the inhibition analog study, therefore 2 mM concentration were used for the inhibition analog study. And inhibitors concentration can be designed according to the obtained Km value, toxicity and solubility. Moreover, similar concentration were used to examine the citrulline transport system in TR-BBB cell lines as well (Reference No. 27).
- L-Type amino acids such as phenylalanine, tyrosine, methionine and citrulline are the substrates of the LAT1 (Table 3) therefore, showing the strong inhibition and others are L-type amino acids showing the stereo-specificity effect because we are using [3H]L-tryptophan for the uptake study. LAT1 is mostly stereo-selective for L-type amino acids.
- In this work, we aimed to examine the transporter involved for the transport of tryptophan therefore, we used various transporter inhibitors/substrates to examine tryptophan inhibition effect. Where we did not evaluated the IC50 values but we examine the competitive inhibition effect or compounds binding site using gabapentin and baclofen (substrates of LAT1) where we found the Ki values 0.21 mM and 1.62 mM, respectively (Figure 2).
- On the other hand, for the pharmacological compounds inhibition (Table 4) effect on tryptophan uptake we had selected concentration of the compounds according to the solubility and toxicity. Verapamil and L-dopa are representing as neuroprotective compounds showing the strong inhibition effect on tryptophan uptake at lower concentration. L-Dopa is a substrate of the LAT1 thus it is showing the strong inhibition effect, whereas verapamil can be transported via ATP-binding cassette transporters (ABC transporters) which have broad field therefore it can show inhibition on the LAT1 substrate transport system as well.
2. The stimulation with LPS and TNF had an only moderate but contrary effects on tryptophan transport in both cell lines. Furthermore, the inhibitors shown to have substantial inhibitory effects in unstimulated cells did not reduce the tryptophan transport but rather increased it. In light of these overall rather confusing data it is not sufficient to come up with speculations in the discussion. Experimental proof of at least some of these speculations is mandatory to increase the trust into the reported data.
Response:
- The LPS and TNF-α have presented opposite results in two cell lines because depending on the cell lines, effect of the inflammatory stimuli can be different. To make our results more consistent, we can correlate L-tryptophan results with the previously published data (PBA & taurine) (Reference No. 29 and 30).
- The inhibitors/compounds (baclofen, L-dopa, gabapentin and verapamil) have shown substantial inhibition on the inhibitory effects (which was done by co-treatment, Table 4). In contrast, for the pretreatment study compounds were used for the 24 h of pretreatment under the inflammatory states. From the co-treatment study, we examine the inhibition study according to that we assume the transporters involvement in the transport study but from the pretreatment study we examine the pharmacological effects of the compounds thus depending on the treatment style (co- and pre-treatment) the effect of the compounds alters the transport mechanism.
- To explain difference co-treatment and pretreatment effect of the compounds on the uptake study, we can see on previously published data (Reference No. 30) which represents the inhibitors of the transporters shown inhibition effect on the inhibition study for 5 min uptake but 24 h of pre-incubation on NSC-34 MT cell lines increased the uptake (showing the pharmacological effects) of the 4-phenylbutyric acid. And we believed that the pretreatment of the neuroprotective compounds can minimize the neuro-inflammatory action induced by the inflammatory stimuli.
3. Inhibition with the siRNA unfortunately reached only 50%. Again only one concentration and one pre-incubation time is reported here. Concentration-dependency and variation of pre-incubation time or pre-incubation design would have been needed. In addition potential cytotoxicity of the siRNA treatment are missing.
Response:
- For the siRNA transfection study, first we used 48 h for siRNA transfection on motor neuronal cell lines which made cell conditions worse/floating (toxicity) therefore, we changed incubation time to 24 h pretreatment, results for neuronal cells lines have shown in our manuscript. On the other hand, 48 h siRNAs treatment was not affected on the endothelial (TR-BBB) cell lines thus, we have mentioned 48 h pretreatment results for the BBB cell lines.
- Moreover, we used siRNA transfection study to examine the transporter expression and involvement of the transporter for the transport of L-tryptophan on the both neuronal and endothelial cell lines, which have shown around 50 % inhibition effects after LAT1/slc7a5 siRNA transfection therefore we did not use siRNAs for various concentrations and pretreatment time on both the cell lines.
- Additionally, we had followed the previously published literatures to perform the siRNA transfection study which are also shown the similar pattern of time and concentration for the transfection study (Ref. 27 and 28).
- SiRNA transfection study we have corrected in the manuscript as well.
Some minor Details:
Figure 1: Statistic Evaluation L- vs. D-Tryptophan missing, would be needed to support statement on stereospecific effect on the uptake (line 154-155)
Response: Statistical difference has been mentioned in Figure 1 and description also changed.
Figure 1: Text for subfigure A and B are mixed up; description of subfigure b is incomplete, meaning of different lines in the plot is missing
Response: Figure 1. Legend description has been corrected as
Figure 1. Stereo-specificity, concentration, and ion dependency uptake of [3H]L-tryptophan in TR-BBB cell lines at 37°C for 5 min at pH 7.4. A) The uptake was carried out in the presence of 2 mM unlabeled L-tryptophan and D-tryptophan. B) Concentration-dependent uptake was performed using unlabeled L-tryptophan concentration ranging from 0-2 mM and the embedded graph shows the Scatchard plot. C) Sodium was replaced with NMG and chloride was replaced with gluconate in the transport buffer. Each point represents the mean ± SEM (n=3-4). *** indicates p < 0.001 significantly different to the control and ### p < 0.001 indicates significantly different to the L-tryptophan.
Figure 5: Scatchard plot too small
Response: Figure 5 has been changed and made bigger sized scatchard plot.
Figure 6: unit of y-axis missing
Response: Figure 6 has been replotted and mentioned in the manuscript.
Additionally to make the same symbolize (TNF-a) we have changed the Figure 7A.
Round 2
Reviewer 2 Report
The authors nicely adopted the manuscript to cover the minor comments. My major comments were, however, essentially limited to direct comments but only very limited modifications in the manuscript. Thus, unfortunately, I see my comments as not adequately addressed.
With respect to the language I can – despite the mentioned review by a native speaker and the provided certificate –not see a substantial improvement. This remains a clear weakness of the manuscript. Here are just a few examples:
Lines 322-323: “According to the published results, transport of neutral amino acids (NAA) by the LAT1 system is sodium independent whereas the basic amino acids are sodium-dependent”
What you likely mean is that the transport of basic amino acids is sodium-dependent. The correct wording for that could be:
“According to the published results, transport of neutral amino acids (NAA) 322 by the LAT1 system is sodium independent whereas that of the basic amino acids are is sodium-dependent”
Lines 335-336: “Taurine, another neuroprotective and transported in the BBB by the taurine transporter (TAUT), choline through the choline transporter (CHT), and acidic amino acids (glutamic acid and aspartic acid), did not affect tryptophan uptake by the TR-BBB cell lines [26,29].”
What I understand is that taurine, Choline and acidic amino acids did not affect the tryptophan uptake. The parts of the sentence which are supposed to further describe taurine and choline properties are not o.k.
Lines 338-340: “In addition to the inhibition, the gabapentin and baclofen are structural analogs of GABA and substrates of LAT1, demonstrate a competitive inhibition effect on the uptake of [3H]L-tryptophan by the TR-BBB cell lines (Figure 2) [19,33].”
Sentence does not make sense. Maybe one could write the sentence in the following way:
“In addition to the inhibition, the Gabapentin and baclofen, are structural analogs of GABA and substrates of LAT1, demonstrate a competitive inhibition effect on the uptake of [3H]L-tryptophan by the TR-BBB cell lines (Figure 2) [19,33].”
Author Response
Response: Thank you so much for your comments and suggestions.
Correction in manuscript are highlighted.
According to the reviewer suggestion, English language and style have been corrected by native English speaker.